# A New Mechanism of Male Plug for Electrical Protection

Rocío Rodríguez [1,*], Manuel Curado [2], Elena Sardiña [1], Jesús Toribio [3]

1 Department of Mechanical Engineering, Catholic University of Ávila, Calle Canteros, s/n, E-05005 Ávila, Spain
2 Escuela Politécnica, Campus Los Jeronimos, s/n, Universidad Católica de Murcia, E-30107 Murcia, Spain
3 Fracture & Structural Integrity Research Group (FSIRG), University of Salamanca (USAL), E.P.S., Campus Viriato, Avda. Requejo 33, 49022 Zamora, Spain
* Correspondence: rocio.rodriguez@ucavila.es

**Abstract:** There are many inventions to avoid the risk of electric contact in the plug. However, the objective of the invention resulting from the research is the proposal of a complementary measure to the existing ones, solving the specific problem of electrocutions that occur in the action of connection and disconnection, due to the contact of the user with the active parts of the plug. The research has focused on the proposal of a design solution that solves the problem of electrocution in these particular and common cases. The objective of this article is the disclosure of a male plug protection mechanism, which has been carried out solely with the use of mechanical methods. Its importance lies in the simplicity and feasibility of the registered utility model as well as in the importance that its implementation may entail for the prevention of serious accidents in the electrical field for domestic and industrial use. The work shows the mechanism and operation of the invention of the proposed male plug in terms of protection against electrical hazards.

**Keywords:** electric protection; male plug; mechanics

## 1. Introduction

The rate of accidents related to occupational injuries is very high in the construction industry compared to any other industry or sector according to the statistics found in the literature [1,2]. The Occupational Safety and Health Administration (OSHA) declared in one of its reports that electrical hazards are in the top four highest rated hazards for the health and safety concerns of the construction industry in the United States. However, the research and development to control and minimize the effects of electrical hazards are being completed efficiently in almost every past decade but the injury cases, either on the field or off the field, are taking place at the same or greater rate. In a report, it is shown that around 51 of the injuries in the field of construction are reported because of the electrical hazards for the year of 2013 [3,4]. So, such alarming figures compel the researchers to be focused more efficiently and effectively to control and overcome the effects of electrical hazards.

Thanks to past research, we now know that electrical hazards are easy to predict and control [4–6]. Nevertheless, there is still the need to test the effectiveness and efficiency of the methods and preventive measures which are being applied to control the rate of electrical hazards. However, there is a need to work on the research and development for the estimation, inspection and assessment of the methods and preventive measures which are being applied to control the rate of electrical hazards. Consequently, to compensate for this gap, the recent research should be able to define and address the problems and issues related to the electrical hazards to ultimately depreciate the rate of injuries and contact of the workers with electrical hazards during their performance.

The electrical injury occurs due to the penetration of electricity in the human body. This penetration or passage of electricity may damage the tissues of the human body. The destruction of tissue only occurs with comparatively larger body currents. With smaller

currents, however, there are already uncontrolled muscle contractions and an influence on the nervous system [7]. This damage is highly dependent of the following factors:

- Current Intensity;
- Current Density;
- Current Path;
- Current Type;
- Current Exposure Duration.

The flow of current through a body may affect the whole body at once or a single part or organ of a body through the tissue damage mechanism. This effect is totally dependent on the above-named factors that play a very important role for a greater or a less damage inside the human body being affected by electricity [8].

The electric current is basically a flow of charges which may be direct or alternating current. The current with one-directional flow of electric charge is called direct current or simply DC. The current with the flow of charges in reverse direction is known as alternating current or simply AC. The hierarchy of controls (HOC) standards are used to simply determine the effectiveness of the protections and preventions involving the evaluation of performances, with a practical approach of risk mitigation [9–11].

The HOC is classified into five basic levels with respect to the measurements, according to the top-to-bottom values of effectiveness. The five classified levels of effectiveness are Elimination, Substitution, Engineering, Administration and Personal Protective Equipment (PPE) in descending order of effectiveness, explaining the HOC for the levels of effectiveness [12].

The first and the most effective level of the control of hazard is Elimination. Elimination can remove all the hazards simultaneously. In the process of Elimination, the whole working phenomenon is changed to permanently remove the hazard.

For example, in the case of electricity sources, the breakage or removal of electrical circuits can remove the hazard altogether. The second level of effectiveness about electrical hazards is the Substitution. In Substitution, the hazard is substituted by another hazard which is not a hazard or is less hazardous to the workers as compared to the previous one, on the place of the work performance. One example includes the usage of a cordless tool which is supported by the power of battery instead of using a corded power tool as it may mitigate hazards related to the electricity cables.

The third level of effective control is Engineering. In engineering services, a protective barrier is formed between the workers and the hazards. This formation of barriers is performed with the utilization of safeguarding technology. For example, the use of rubber or polymer-coated instruments such as spanners or screw drivers which may not affect the workers in response to the electrical current. Moreover, the use of an insulator ladder may also create a barrier between the worker and the electricity power lines or ultimately the electrical hazard.

The fourth level of effective control is Administration, which implements the rules, policies, regulations and standards of operations as well. The scheduling at the workplace is also a part of administration work. For example, the implementation of the lockout standard of operation can reduce the rate of workers being shocked by the electrical sources or hazards. The level of administration should be strong enough to implement all of the rules and policies as strong administration can make an organization stick to rules and discipline [12–15].

The fifth and last level of effective control is Personal Protective Equipment. This level of control directly fixes the equipment of safety and protection on the bodies or workers during their performances at the workplace. Some examples of personal protective equipment or (PPEs) are ear plugs, goggles, helmets, sleeves, gloves and respirators. The use of these mini but effective equipment is very important, as these small equipment may help the workers to keep safe with greater effect from electrical or any other hazard. The use of PPEs is increasing day by day, and a number of training sessions are also organized to make people aware of their existence and usage [16].

In order to prevent deaths by electrocution caused by direct contact with plugs, both at the domestic and industrial level, one of the measures adopted is the covering of the active parts of the installation, those through which the electrical current circulates. Currently, inventions have been registered on new electrical connectors and electrical connection systems [17] as well as new socket insulation [18], in which different solutions are proposed to avoid electrical risks. One of the most recorded types of accidents occurs specifically in the action of connecting and disconnecting the plug due to the direct contact of the person with the terminals of the male plug, which, once connected, are active parts [3].

In the proposed study, it is decided to improve the plug model "male part" or plug in order to significantly reduce the risk of direct contact and thus the possible electrical accident. There are several plug manufacturers who have opted for the semi-coverage of the terminals of said plugs. Some of the examples of current existing models are semi-coatings on terminals with insulating material and mechanisms that avoid direct contact in the female part of the plug such as utility models [19,20]. This measure reduces the risk of contact in the plug whilst plugging in the device, but it does not completely prevent it. In the plug model, a complete covering with a retractable mechanism is proposed, which adapts to the position of the plug at all times during the action of connection and disconnection with the plug or "female part".

The main advantage it provides with respect to current plug protection systems is the isolation of the active parts during the entire process of connecting and disconnecting machines to the electrical network. The problem solved by the design is the likely electrocution by direct contact with the terminals of a plug's male plug, which, when connected or disconnected, has uncovered areas where the electric current circulates, being susceptible to contact with people, animals or objects; this can in turn transmit electricity through them, producing possible electrical discharges. The proposed mechanism, easy to manufacture and handle due to its mechanical and automatic drive, makes its implementation safe, low cost and with a minimal economic impact on the end user.

## 2. Materials and Methods

The sector in which the invention is framed is the manufacture of electrical mechanisms, more specifically in the manufacture of male plugs, through which most appliances, machines and appliances that run on electric power are connected to the grid, just by inserting this plug into a base or outlet of any kind.

Such devices are well known in the state of the art. A male plug or plug is a piece of insulating material from which at least two rods called metal nipples that are inserted into the female plug stand out to establish the electrical connection to the outlet. Nowadays, most plugs are recessed, so that the insulating part of the male plug is also introduced into the power outlet, and therefore, there is hardly the possibility of grabbing the nipples while connected to the current and therefore receiving an electric shock while it is being handled; there is still a risk though that children, who have smaller fingers, are able to access the connection.

In the state of the art, there are many studies about the different properties of mechanisms for the protection of male nipples [21]. However, there is no mechanism similar to our proposal. This model describes a pin in which the nipple assembly is inside an insulating envelope protruding from the plug body and is equipped with a retractable movement when the connection is established by hiding inside the plug body and protruding when the plug is disconnected by springs arranged at the base where the envelope is housed. The drawback of this device is that it offers no guarantee that the nipples will still not be touched by hand, since the envelope leaves a gap with a diameter similar to that of the pin.

Based on the above technique, one of the aims of the present invention is to provide a male plug with individual protection of all its nipples, or at least from the phase nipples since neutral or ground poses no danger; so that even if there is manual contact with any of them, when connected or half connected, it is not a direct contact on the metal part of the nipple, but on an insulating coating and therefore an electrical cramp cannot occur when handled.

In order to achieve the above-mentioned objectives, the invention proposes a male plug with anti-contact protection, which has the characteristics described in Section 3. More specifically, as indicated in the previous point, the main feature is that each of the metal nipples that establish the connection has an individual means of protection, and therefore, it is not possible that there is contact with it at any time, either when it is unplugged, or plugged-in, nor the connection to medium establish with the nipples partially inserted into the base or outlet.

*2.1. Material Selection*

The selection of material is considered as a very crucial step because even with a very minor mistake, it may lead toward the failure of the product. The materials that are considered the most for this particularly designed applications are thermoset polymers. Thermosetting polymers are highly insulated plastics with greater heat stability. These plastics undergo a chemical reaction when heat is supplied to them. These chemical reactions enhance the rate of cross linking between the polymeric chains, causing the curing of plastics. These types of electrical applications require basic properties; the first is insulation and the second one is greater heat stability. The fluctuations in the flow of electricity may cause ignition, producing heat. So, in this case, the heat stability of material is very important. On the other hand, to carry or operate with these electrical switches, sockets or wirings, insulated materials are preferred. The examples of thermosetting plastic are urea–formaldehyde, polyimide, melamine formaldehyde, vinyl ester and polyester. Among all of these materials, the urea–formaldehyde is most suitable for this application. The insulation of urea–formaldehyde is excellent as well, as it shows greater tensile strength and flexural modulus with greater heat stability and heat distortion temperature. The toughness of urea–formaldehyde is high and its fire-retarding properties are also excellent, making it a suitable material for electrical fittings and plugs-based applications. The pure form of urea–formaldehyde shows the strength up to 5 GPa [22]. However, its toughness may be optimized as per the requirement of the application . While on the other hand, the polyimides have low impact strength, although it has tremendous tensile and compression strength even for a wide range of temperatures from around −270 to 300 °C. However, polyimide is an expensive material that could be a hurdle in the commercialization of this product [23,24]. The pure form of polyester provides the strength up to the value of around 90 MPa. However, in comparison with urea–formaldehyde, its temerature range is very low, which makes it unsuitable to be used in fire-retardant and high-temperature applications [25–27]. The urea–formaldehyde shows mechanical stablity with strength of up to 5 GPa, while the polyimide and polyester show the strength from 60 to 90 MPa. However, the thermal stability of polyimide is a bit higher, but it shows low impact strength as compared to urea–formaldehyde. While, on the other hand, polyester has low strength and flame retardancy which are very important for this particuar application.

*2.2. Process Selection and Optimization*

The selection process for the molding of these electrical plugs should be optimized as per the requirements of the final application. Urea–formaldehyde can be processed or molded using injection molding or compression molding. However, the quality of injection molding products can be optimized in a wide range as compared to the compression molding. The optimization of the product quality can be seen many times in the literature through simulation software. The commonly used software are Moldflow and Solidworks Plastics. The simulation is performed considering the multiple factors such as injection point, mold temperature, melt temperature, injection pressure, machine RPM and type of material. The simulation allows optimizing parameters for the best quality products under specified conditions. Mostly, the optimized thickness for electrical plugs and fittings are observed at the value of 1.2 to 2.5 mm, which may depend on the material, processing parameters and type of application. To summarize, urea–formaldehyde is observed to be a great material for electrical plugs, switches and fittings.

## 3. Results

As can be seen in the referenced Figure 1a,b, the male plug with the contact protection object of this model has an insulating envelope (1), inside which the connection of at least two-phase cables (3) with the corresponding metal nipples (2) are established, which when inserted into the terminals of the base (6) establish the connection.

According to the characteristics of the invention, this plug has an insulating envelope (4) that protrudes from the body of the insulating envelope (1) until the corresponding nipple (2) is fully covered, which is driven and maintained in this position by means of a spring (5) arranged in an existing hole in the insulating envelope (1) behind the corresponding nipple (2) as long as the plug does not attempt to connect to a base (6) (see Figure 1a). When the plug is inserted into the base (6) to make the connection, the envelope retracts and enters the body of the plug, compressing the aforementioned spring (5) (see Figure 1b).

To complement the description that is being made and in order to facilitate the understanding of the characteristics of the invention, a set of drawings support the descriptive memory for illustrative and non-limiting purposes (Figure 1).

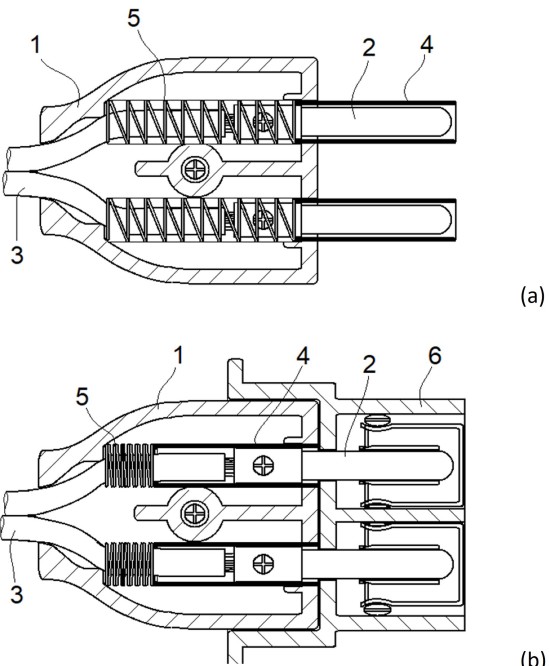

**Figure 1.** The utility model (**a**) represents a male plug when not connected. (**b**) represents the previous plug when inserted and connected to a base outlet.

## 4. Discussion

### 4.1. General Instructions for Safety

As a general rule, the hierarchy of controls works in such a way that the control or preventive measures at higher scale are found to be more efficient, effective and protective, but they require a huge effort for their implementations. As explained in Figure 1, the top three levels of control are considered as technological controls, while the bottom two controls are considered as the controls of behavior addressing the body language of the people during work at the workplace and have to be changed if it is not as per the standards of operations. The definition of these controls according to the United States Centers for Disease Control and Prevention is as follows [12–15].

■ The elimination and substitution are at the top to decrease the effects of hazards as they are highly effective. The implementation of these two levels is quite difficult,

especially in an existing working process. However, they are not so expensive nor difficult to be implemented during the design stage (i.e., the beginning of the process).

■　The controls based on the engineering are quite tricky in terms of design, but if they are designed in a great manner and according to the requirements of the process, they are highly efficient and effective for the protection of the workers from the hazards. They are generally independent of contact with workers. The cost of the engineering controls is higher but in the long term, the operating cost is quite low as compared to the starting one. Furthermore, this smart and timely investment on engineering controls can reduce the operating cost to the point where savings or profit can be generated in great amounts.

■　The controls of administration and personal protection equipment (PPE) are used largely in the fields with greater processes but are considered as less efficient, effective and, very importantly, less protective as compared to the other effective control measures. The initial expense of these control measures is low, but in the long term, the cost of their sustainability can increase up to a very high value. Moreover, the involvement of workers in these control measures is very high, as the workers have to put in a greater effort in these control measures for a poorer hazard control.

### 4.2. Causes of Electrical Injuries by Contact with the Male Plug Terminal

There are a number of factors which may cause an electrical hazard or accident ultimately causing the electrical injuries. These electrical injuries may occur because of two phenomena. These two are Resistive Coupling and Capacitive Coupling.

#### 4.2.1. Resistive Coupling

In this phenomenon, the body makes direct contact with the electrical sources and earth such as a live wire or a socket touching by mistake. These electrical sources may be in the machinery due to the faults in manufacturing or leakage currents. The leakage current is produced when the potential of the earth becomes less than the potential of the machinery or equipment which is being under usage. The second possible condition for leakage current is when possible earth potentials exist. In this case, a very minute value of applied voltage or current may result in a greater electrical injury. This minor value of current is known as micro-shock. In other words, the micro-shock is a micro-scale current which lies in the value ranges from 50 to 100 mA. This amount of current can even cause a severe electrical injury when applied to a certain body part such as the human heart through the line of central veins, where it may result as ventricular fibrillation [28].

#### 4.2.2. Capacitive Coupling

In the capacitive coupling, the body makes contact with earth and the electrical source and acts as a plate of a capacitor. The capacitor is a combination of two plates with an insulator between them in order to store charge ultimately. For example a tube light of 50 Hz frequency taken apart from the human body by an insulator (generally air is taken) may be considered as one plate, while the human body can play a role of another capacitor plate. It is known that the alternating current can pass through the capacitor, inducing current in the human body being preyed by the electrical shock or current.

In the case of the magnetic resonance suite of imaging, the strong variations in the magnetic field can induce a strong shock in the body of the worker or patient. This induction of voltage may affect the pulse rate and the saturation of oxygen as well; this may be recorded on digital pulse and oxygen meters. However, on a serious note, it may also cause skin burns and localized heating as well. In the literature, there are cases available in which the loss of fingers is recorded because of the very complex and severe burns [28]. There are very severe effects that are a consequence of the interaction between the human body and electrical currents. Some of the major effects are described below with the ranges of current values:

•　1–5 mA causes tingling pain;

- 5–15 mA causes pain;
- 15–50 mA causes contraction of tonic muscle;
- 50–70 mA causes respiratory arrest;
- 70–100 mA causes contraction of respiratory muscle;
- >100 mA causes ventricular fibrillation.

## 5. Conclusions

In general terms, the electrical equipment is maintained regularly. The safety of the worker, through not being in contact with the electrical items, and the use of an antistatic worker's kit, is also covered in general measures.

Specific measures include the adequate conservation of equivalent power and the utilization of circuit breakers and floating circuits. In equipment design, specific codes and standards are kept under consideration to design the electrical process with complete optimization, minimizing the effects of any type of instability and misfortune during the process, ultimately lowering the chances of electrical injuries.

Keeping under consideration the electrical hazards in residential wirings, especially in the switch, plug and socket boards in the houses, there are a number of cases reported and addressed in the literature. The causes explained for the occurrence of these hazards were a lack of interest or carelessness. The most affected prey of electrical hazards in a domestic context are children. This research and the suggested model allows the body to establish a safe and protective contact with the electrical wiring or sources.

The great protection from the electrical switches and plugs can be achieved when a better and safer contact is ensured by the design of the plug or socket, which is usually made of plastic or any other insulator material. These front covers or knobs are touched mostly by children or pets during their use.

The biggest disadvantage of using these devices is that they are covered with knobs or plastic covers. This means that in order to plug in or switch on these devices, the removal of covers is necessary and therefore, the direct contact of fingers or the human body is possible. It is very important then that a very quick and timely response takes place or that the fingers are pulled out quickly. This increases the chances of hazard dominantly. Apart from this, in open conditions in a house, there are very high chances that a child will try to touch the plug, thus increasing the probability of electrical hazard.

The main objective of this modeling and relevant research is to ensure the protection of all of the plug nipples especially from each other, without lessening the efficiency of the earth or grounded nipple. The use of this protective core does not allow the current to pass through it, as they are made of insulating material. This insulating material does not even allow contact between the nipples of the plug in order to avoid the electrocution, which is a very dominant advantage of this model.

In order to accomplish the aims of this research effort, the following proposal is presented: to develop protection covers for each and every single plug nipple. This will prevent contact between the nipples, as every nipple will have its own protection case. This model exhibits a very strong and effective safety and protection for the human body that is going to be in contact with this plug, which is an excellent and crucial feature of this electrical plug model.

## 6. Patents

This utility model of a simple male plug protection mechanism has been registered with the following Code corresponding to the invention's country of origin, Spain: U202030547.

**Author Contributions:** All authors contributed equally to this work. All authors have read and agreed to the published version of the manuscript.

**Funding:** This research received no external funding.

**Informed Consent Statement:** Not applicable.

**Conflicts of Interest:** The authors declare no conflict of interest.

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
