# Peer review of "A New Mechanism of Male Plug for Electrical Protection"

_inventions, doi:10.3390/inventions7040123_

Round 1

Reviewer 1 Report

The article of a new mechanism of male plug for electrical protection has technical sound. However, there are require to update the article.

Abstract: too shot and unclear in term of method, and finding.

The invented model is unclear and should clearly explain in detail on the design etc.

The finding is not clearly tested. (experimental verification etc)

References; Most of references are outdated.

Author Response

We are grateful for the review.

These suggestions have been included in the revised version of the manuscript.

The references have been updated as far as possible, the authors hope that the reviewer will take into account that the problem of the prevention of electrical hazards based on proposals for plug protection mechanisms is a topic of current relevance, but it is an ancestral problem and therefore the major findings and uses have not been made recently.

The authors understand and appreciate your comments on the need to know the details of the design as well as the testing of the prototype. However, it is not possible to present the specific characteristics or details of the prototype as they are industrial secrets of the producing companies that are currently in the exploitation phase of the utility model and therefore, the authors are not authorised as inventors to publish them. An example of the same problem, given the industrial nature and novelty of the invention, is one of our contributions to this journal; Ear Canal Cleaning Mechanism (https://doi.org/10.3390/inventions7010020) in which it has not been possible to describe the mechanism with specific but general characteristics as in this article.

Reviewer 2 Report

Thanks to the authors for preparing the paper!
Please see your PDF document with my comments! A few spelling mistakes need to be corrected.

There is also the question of the novelty of the protected electrical plug, because these insulating envelopes have been used for a long time, e.g. in laboratory cable plugs.

Author Response

We are grateful for the review.

These suggestions have been included in the revised version of the manuscript.

This article studies a new possible solution from the point of view of the male plug’s protection, with a clear technical contribution in risk prevention, which has been registered as a utility model in the official register of Spain’s intellectual property, U0174296 according to the registry of the Spanish patent office. This registration guarantees the novelty of the invention with respect to all mechanisms for this purpose proposed to date, which of course includes the protection of the laboratory sockets proposed above.

Reviewer 3 Report

The paper introduces an original idea that may lead to a new mechanism of male plug for electrical protection. 

The entire patent proposal is presented and discussed in only one figure.

Even the theoretical proposal looks promissing and meritoriuos, why is not presented an experimental ptototype of the discussed male plug (completed by adequate measurements) that my suport sufficiently the proposed patent?

Only to introduce a theoretical presentation of the idea looks not full enough to convince about the feasibility of the whole proposal.

The above inconvenience looks to be the main drawback of the presented paper.

Author Response

We are grateful for the review.

The authors understand and appreciate your comments on the need to know the details of the design as well as the testing of the prototype. However, it is not possible to present the specific characteristics or details of the prototype as they are industrial secrets of the producing companies that are currently in the exploitation phase of the utility model and therefore, the authors are not authorised as inventors to publish them. An example of the same problem, given the industrial nature and novelty of the invention, is one of our contributions to this journal; Ear Canal Cleaning Mechanism (https://doi.org/10.3390/inventions7010020) in which it has not been possible to describe the mechanism with specific but general characteristics as in this article.

Round 2

Reviewer 1 Report

This paper seem unclear in term of method. Suggested to improve the  methodology in term of process flow and detail explanation. 

Reviewer 3 Report

The manuscript is proposed for journal publication.